# Novel transgenic pigs with enhanced growth and reduced environmental impact

Xianwei Zhang[1,2†], Zicong Li[1†], Huaqiang Yang[1,2], Dewu Liu[1], Gengyuan Cai[1,2], Guoling Li[1], Jianxin Mo[1], Dehua Wang[1], Cuili Zhong[1], Haoqiang Wang[1], Yue Sun[1], Junsong Shi[2], Enqin Zheng[1], Fanming Meng[1], Mao Zhang[1], Xiaoyan He[1,2], Rong Zhou[2], Jian Zhang[2], Miaorong Huang[1], Ran Zhang[3], Ning Li[3], Mingzhe Fan[4], Jinzeng Yang[5], Zhenfang Wu[1,2]*

[1]College of Animal Science, South China Agricultural University, Guangzhou, China; [2]National Engineering Research Center for Breeding Swine Industry, Guangdong Wens Foodstuff Group Co., Ltd, Yunfu, China; [3]College of Biological Science, China Agricultural University, Beijing, China; [4]Department of Animal Biosciences, University of Guelph, Guelph, Canada; [5]Department of Human Nutrition, Food and Animal Sciences, University of Hawaii at Manoa, Honolulu, United States

**Abstract** In pig production, inefficient feed digestion causes excessive nutrients such as phosphorus and nitrogen to be released to the environment. To address the issue of environmental emissions, we established transgenic pigs harboring a single-copy quad-cistronic transgene and simultaneously expressing three microbial enzymes, β-glucanase, xylanase, and phytase in the salivary glands. All the transgenic enzymes were successfully expressed, and the digestion of non-starch polysaccharides (NSPs) and phytate in the feedstuff was enhanced. Fecal nitrogen and phosphorus outputs in the transgenic pigs were reduced by 23.2–45.8%, and growth rate improved by 23.0% (gilts) and 24.4% (boars) compared with that of age-matched wild-type littermates under the same dietary treatment. The transgenic pigs showed an 11.5–14.5% improvement in feed conversion rate compared with the wild-type pigs. These findings indicate that the transgenic pigs are promising resources for improving feed efficiency and reducing environmental impact.
DOI: https://doi.org/10.7554/eLife.34286.001

*For correspondence:
wzfemail@163.com

†These authors contributed equally to this work

Competing interests: The authors declare that no competing interests exist.

## Introduction

Annual global pig production is approximately 1.2 billion heads, with more than half produced in China (**USDA, 2016**). Grains are the main feedstuff of the pig industry; however, its production capacity in China and many other countries is insufficient. The livestock industry often allows animals to achieve maximal growth in order to fully utilize their economic outputs, yet inefficient feed digestion can cause serious nutrient emissions to the environment. Two nutrients that have received the most attention from environmental groups are nitrogen (N) and phosphorus (P) which are often supplied in excessive amounts in the diet in order to ensure maximal growth. In pig production, only 1/3 of feed N and P is metabolically utilized from cereal and soybean-based diets. The deposition rate of N is only 25–32% in grower-finisher pigs (**Shirali et al., 2012**). It has been reported that N excretion is up to 20 kg/sow/year and 25 kg/boar/year (**DEFRA, 2007**). Approximately 51% of N intake is excreted in urine, which is mainly from protein metabolism and underutilized amino acids and nonprotein nitrogen (NPN) (**Shirali et al., 2012**). Fecal N excretion comes from undigested protein fractions and endogenous tissue losses such as digestive enzyme secretions and desquamation of intestinal cells, which accounts for 17% of the N intake (**Dourmad et al., 1999**). Only approximately 30%

**eLife digest** The bodily waste that pigs produce contains high levels of chemicals that can damage the environment, such as nitrogen and phosphorus. For example, when excessive amounts of these two compounds make their way into the water, they can cause blue-green algae to grow too much, which asphyxiates other life in the water.

Pigs produce a lot of nitrogen and phosphorus because they cannot efficiently digest their food. In particular, the animals lack the enzymes required to break down two types of molecules present in their feedstuff: phytates and non-starch polysaccharides (NSPs).

Zhang, Li et al. take four microbial genes which code for the enzymes needed to digest NSPs and phytates, and they add these DNA sequences into the genomes of pigs. The animals then produce enzymes in their saliva that transform NSPs and phytates into molecules which can be used by their digestive system. The pigs thus get more energy from their food, and they grow faster and bigger. They also produce less nitrogen and phosphorus in their waste.

Over 1.2 billion pigs are farmed each year, and they are the most economically important meat source in the world. Raising animals that can digest their food better would reduce the need for pig feed, increase productivity and reduce environmental pollution. However, discussions with policy makers and with the public will be necessary before these results can be adopted by the farming industry.

DOI: https://doi.org/10.7554/eLife.34286.002

of P is retained in a grower-finisher pig on a cereal-soybean meal-based diet. In total, 70% of ingested P is excreted either through the feces or urine (*Dourmad et al., 1999*). The N and P from animal excreta pollute the water, soil, or air of intensive pig production sites (*Osada et al., 2011*; *Philippe et al., 2011*; *Carter and Kim, 2013*). Surface water becomes eutrophic following excessive P and N inputs, thereby causing overgrowth of blue-green algae and death of aquatic animals (*Jongbloed and Lenis, 1998*; *Poulsen, 2000*). Considering all these aspects, improving nutrient utilization in feed is of great significance to maximize feed grain utilization as well as for environmental conservation.

Non-starch polysaccharides (NSPs) are primarily present in plant cell walls (*McDougall et al., 1996*; *Sarkar et al., 2009*). In cereal grains, arabinoxylans and β-glucans are found in the cell walls of the protein-rich aleurone layer and starchy endosperm and can act as a barrier to nutrient hydrolysis and absorption (*Bacic and Stone, 1981*). Similarly, the cell wall polysaccharides of soybean, canola seed, and peas may also be responsible for this nutrient-encapsulating effect (*Omogbenigun et al., 2004*). Therefore, NSPs are the main anti-nutrient factors of cereal and bran (*Fangel et al., 2012*; *Sarkar et al., 2009*). Due to a lack of endogenous NSP-degrading enzymes (NSPases), pigs are inherently incapable of digesting NSPs (*Hooda et al., 2010*), but can partially degrade this material through the action of the natural microbial community in their intestinal tract. High-P emission from monogastric animals such as pigs and poultry arises from their poor physiological ability to hydrolyze plant phytates, which account for up to 80% of P in common cereal grains, oil seed meals, and by-products (*Ravindran et al., 1994*). Phytates are negatively charged saturated cyclic acids that can bind to positively charged molecules in the diet such as minerals and protein, thereby reducing nutrient digestibility and increasing discharge of the unabsorbed nutrients to the environment (*Dersjant-Li et al., 2015*).

Various methods have been employed to address the issues of inefficient utilization of feed nutrients in the pig industry. For example, dietary supplementation of phytate- or NSP-degrading enzymes has been proposed to reduce P or N emissions from pig farms (*Kiarie et al., 2010*; *Zijlstra et al., 2010*), as well as increasing pig body weight gain and feed conversion efficiency (*Diebold et al., 2004*; *Willamil et al., 2012*; *Woyengo and Nyachoti, 2011*). Recent advancements in genetic engineering and animal cloning technologies have facilitated in the establishment of genetically modified pigs with economically significant traits. Transgenic (TG) pig lines secreting salivary bacterial phytases have been generated previously. The P content of fecal matter from TG weaner and grower-finisher pigs fed on soybean meals was decreased by as much as 75% and 56%, respectively, compared to their non-TG counterparts. Endogenous salivary phytase significantly

promoted the digestion of P from dietary phytates (*Golovan et al., 2001*). To our knowledge, no pig lines that express multi-NSP-degrading enzymes have been established to date. Here, we established stable transgenic pig lines that co-expressed NSP-degrading enzymes (β-glucanase and xylanase) and phytase in saliva. Multiple enzymes coordinately degrade NSPs and phytates in feed grains. We also report the grain digestibility, nutrient emission, growth performances, and feed conversion rate of the TG pigs compared with their wild-type littermates.

## Results

### Optimization and construction of a 2A-mediated salivary gland-specific multi-transgene

Through characterization of multiple codon-optimized β-glucanase genes fused with the N-terminal porcine parotid secretory protein (PSP) signal peptide, we have determined that *Bispora sp. MEY-1* endo-β-glucanase from (BG17A) and *Bacillus licheniformis*β−1,3–1,4-glucanase (EG1314) exhibited optimal activity and stability in porcine cells, and the pH condition is compatible to that of the pig digestive tract (*Zhang et al., 2015*). We previously reported that the fused BG17A and EG1314, which was linked by a self-cleaving 2A peptide, had a broader optimal pH range and higher stability in an acidic environment than either of them alone (*Zhang et al., 2015*). After codon optimization and fusion with the pig PSP signal peptide, three xylanases (XYNB, XYL11, and XYF63 (also known as XYN11F63)) were transfected into PK15 cells and subjected to enzymatic activity assay. Among the three xylanases, XYNB presented the highest enzymatic activity (*Figure 1—figure supplement 1A*) and stability (*Figure 1—figure supplement 1B*). In addition, XYNB showed greater resistance to peptic and tryptic hydrolysis than the other two xylanases (*Figure 1—figure supplement 1C–E*). As for the two phytases, *Citrobacter freundii* APPA (CAPPA) only had two narrow peaks at the optimal pH levels of 2.5 and 5.0, respectively (*Figure 1—figure supplement 2A*), whereas *Escherichia coli* APPA (EAPPA) exhibited a broad optimal pH ranging from 1.5 to 5.0 (*Figure 1—figure supplement 2B*). EAPPA was more tolerant of pepsin and trypsin than CAPPA. There was almost no reduction in activity of EAPPA after a 2 hr pepsin treatment, whereas 52.2% of the biological activity was left for CAPPA after treatment (*Figure 1—figure supplement 2C*). When treated with trypsin alone, EAPPA and CAPPA retained 98.2% and 39.7% of their activity, respectively; when treated with trypsin +EDTA, EAPPA and CAPPA retained 31.8% and 13.7% of their activity, respectively (*Figure 1—figure supplement 2D*). Based on these results, two β-glucanases genes (*bg17A* and *eg1314*), a xylanase gene (*xynB*), and a phytase gene (*eappA*) showed better performance that the other candidate transgenes.

A polycistronic cassette of fusion enzymes was constructed by head-to-tail ligation of four selected genes and flanked on the 3′ end by an Hemagglutinin (HA) tag. The DNA sequences of the self-cleaving peptides E2A, T2A, and P2A were used as linkers between the coding DNA sequences of two neighboring enzymes (*Figure 1—figure supplement 3A*). Before construction of the final TG vector, the fusion enzyme sequences were ligated downstream of the CMV promoter, and the expression level and enzymatic activity of each fusion enzyme were measured in porcine cells. We were able to detect the expression of all the four active enzymes in the PK15 cells, although the expression level and enzymatic activity of each recombinant fusion enzyme was lower than its original monomeric counterpart (*Figure 1—figure supplement 3B and C*). The lower transfection efficiency of the large transgene construct likely accounted for the observed lower expression and enzymatic activity in cells. We then employed a mouse PSP promoter to replace the CMV promoter to control the salivary gland-specific expression of the fusion gene, and this expression cassette was inserted, together with a CMV promoter-driven neo-EGFP fusion gene, into the piggyBac transposon vector to form a TG vector, namely, pPB-mPSP-*BgEgXyAp-neoGFP* (*Figure 1A*).

### Generation of TG pigs

The resulting TG piggyBac transposon vector (pPB-mPSP-*BgEgXyAp-neoGFP*) and a piggyBac transposase expression vector (*hyPBase*) were co-transfected into the porcine fetal fibroblasts (PFFs) of a male Duroc pig. Transfected PFFs were selected with G418 for approximately two weeks, and the resulting EGFP-expressing cell colonies were pooled and identified by PCR for the presence of the transgene. Four colonies with great EGFP expression were used as donor cells for somatic cell

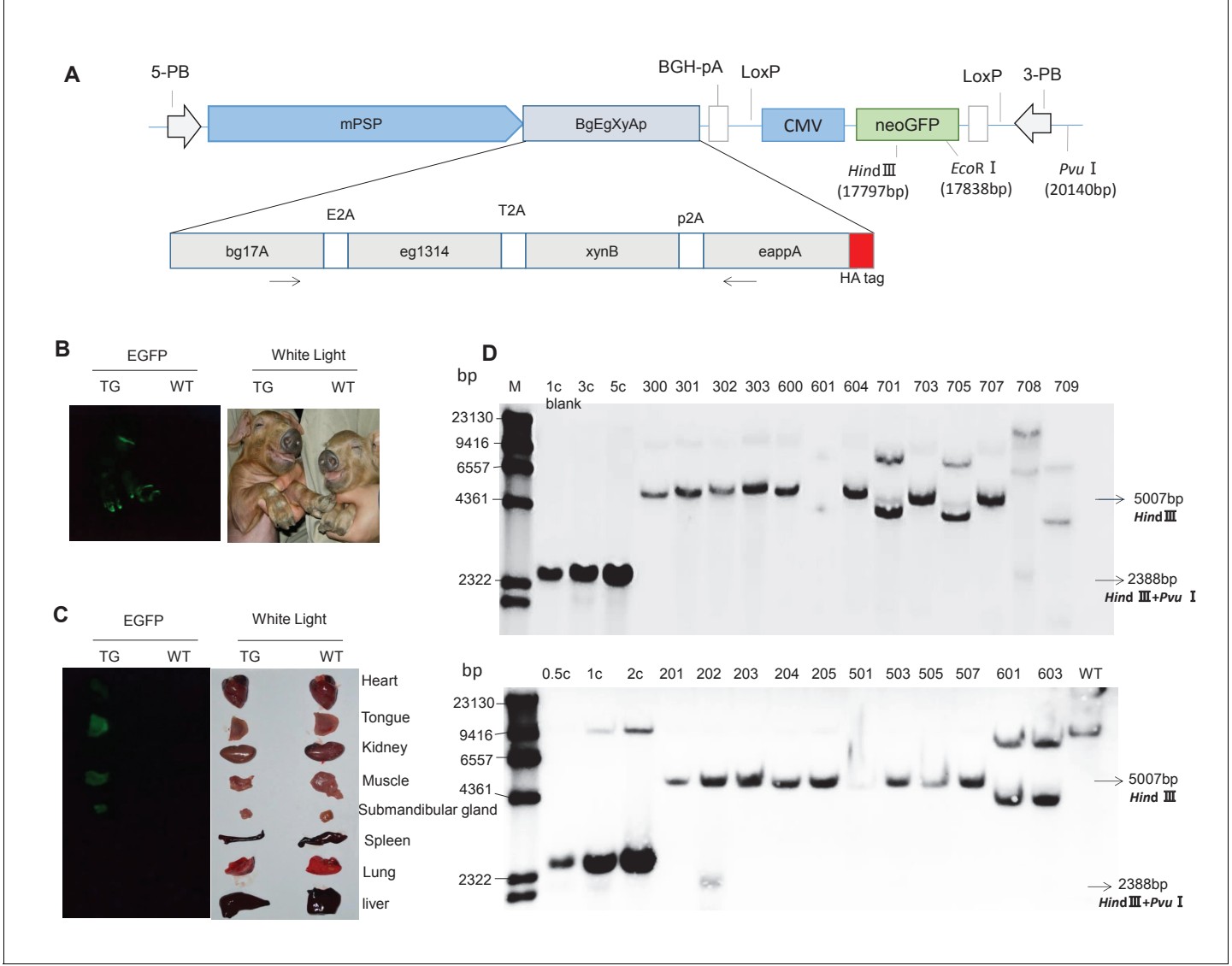

**Figure 1.** Presence of the transgene in TG pigs. (**A**) DNA construct that was integrated into the pig genome for expression of the transgenic fusion enzyme in saliva. mPSP: Mouse parotid secretory protein promoter. BGH: Bovine growth hormone polyadenylation signal. Total length: 19,886 bp. (**B**) Expression of EGFP in the whole body of TG pigs. (**C**) Expression of EGFP in the heart, tongue, kidney, muscle, submandibular gland, spleen, lung, and liver of TG pigs. (**D**) Southern blot analysis of multi-enzyme transgene integration in TG pigs. 0.5 c, 1 c, 2 c, 3 c, and 5 c represent copy number of transgenic vector used as loading controls. The probe is shown in *Figure 1A*. Blank: Blank control (ddH$_2$O).

DOI: https://doi.org/10.7554/eLife.34286.003

The following figure supplements are available for figure 1:

**Figure supplement 1.** Characterization of three xylanases (XYNB, XYL11, and XYN63) expressed in the PK15 cells.

DOI: https://doi.org/10.7554/eLife.34286.004

**Figure supplement 2.** Characterization of two phytases (CAPPA and EAPPA) expressed in the PK15 cells.

DOI: https://doi.org/10.7554/eLife.34286.005

**Figure supplement 3.** Construction of a *BgEgXyAp* polycistron and its enzymatic activity in PK15 cells.

DOI: https://doi.org/10.7554/eLife.34286.006

**Figure supplement 4.** Genotyping analysis of founder TG pigs.

DOI: https://doi.org/10.7554/eLife.34286.007

**Figure supplement 5.** Determination of the linked transgene (*BgEgXyAp*) copy number in the transgenic (TG) founders *vs* wild (WT) pigs.

DOI: https://doi.org/10.7554/eLife.34286.008

nuclear transfer. A total of 4008 reconstructed embryos were generated and transferred to 16 recipient sows (*Supplementary file 11*). Thirty-three live and two stillborn cloned piglets were born, of which 25 founders were positive for transgene by PCR detection (*Figure 1—figure supplement 4A and B*). Bright green fluorescence signals were observed in hoof, tongue, heart, muscle, and submandibular gland (*Figure 1B and C*). Among the 25 TG founders, five piglets (601, 603, 701, 705, and 709) harbored the fragments of the ampicillin-resistance gene of the transgene vector (*Figure 1—figure supplement 4A*), implying the occurrence of a random, but not transposon-mediated transgene integration into host cell genome. The other 20 piglets harbored the intact transgene expression cassette (total length: 19,886 bp). Southern blotting, quantitative PCR, inverse PCR, and sequencing results further demonstrated that 19 piglets carried a single copy of the transgene (*Figure 1D*; *Figure 1—figure supplement 4C*; *Figure 1—figure supplement 5*; *Supplementary file 1*), of which two carried a single copy of the transgene that was inserted into intron 1 of *Legumain* (line 1) (*Figure 1—figure supplement 4C*; *Figure 1—figure supplement 5*; *Supplementary file 1*), 17 carried a single copy of the transgene that was integrated into intron 5 of *CEP112* (line 2, of which eight piglets survived) (*Figure 1D*; *Figure 1—figure supplement 5*; *Supplementary file 1*). One (708) carried three copies of the transgene, and the integration site was the intergenic region between *LOC100525528* and *CXCL2* (*Figure 1D*; *Supplementary file 1*). There were a total of 16 piglets survived to weaning. Thirteen of them are positive for transgene, including nine piggyBac-mediated (one of line 1 and 8 of line 2) and four randomly integrated transgenic pigs. Eight of the transgenic pigs survived to sexual maturity.

RT-PCR analysis indicated that the *BgEgXyAp*, bg17, eg1314, xynB, and eappA transgenes were unambiguously expressed in the parotid, submandibular, and sublingual glands, whereas these were undetectable in the other tissues of the TG founders such as the lungs, heart, liver, stomach, spleen, kidney, duodenum, colon, and muscle (*Figure 2A*; *Figure 2—figure supplement 1A*). Quantitative PCR analysis indicated that the highest *BgEgXyAp* transgene expression levels were observed in the parotid gland, followed by the submandibular and sublingual glands, and trace or undetectable level were observed in the other tissues of the TG founders (*Figure 2—figure supplement 1B*). Ectopic expression was not observed in this study. Western blot analysis demonstrated the expression of β-glucanase, xylanase, and phytase in the saliva of the TG founders (*Figure 2B*). During the feeding period, the TG pigs (line 1 and line 2) produced 0.3–2.3 U/mg of β-glucanase, 0.6–2.4 U/mg of xylanase, and 0.5–5.7 U/mg of phytase in the saliva (*Figure 2C–E*). The total salivary protein concentrations of the TG and WT pigs are shown in *Figure 2F*.

Pigs of TG line two that harbored a single-copy transgene within *CEP112* intron were used in growth trials and feed evaluations. The TG line two pigs were crossed with WT Duroc pigs, which generated 116 F1 progeny, of which 57 tested positive for transgene. Furthermore, 404 of the F2 progeny were sired, of which 231 were positive for transgene.

## Measurement of enzyme production in TG pigs

To understand the effect of the three enzymes of the TG pigs on nutrient digestion, we investigated the pattern of salivary secretion and enzyme production in the TG pigs. Saliva was collected from the unilateral parotid glands of the TG pigs and analyzed in terms of enzyme yield. The average β-glucanase, xylanase, and phytase yields were 2,331.8, 2,413.4, and 2,935.2 U per kilogram meal, respectively, in grower pigs, and 920.8, 939.0, and 1,042.2 U per kilogram meal, respectively, in finisher pigs (*Table 1*). The volume of saliva collected from the parotid gland of the finisher pigs was significantly lower compared to that of the grower pigs, which may be attributable to the shorter feeding time (Ft) in finisher pigs. Furthermore, saliva and enzyme production at different time points were evaluated. The results show that the pigs only secreted saliva from parotid gland during Ft. At the other time points, including 10 min before or after feeding (Bf or Af), insignificant amount of saliva was secreted (*Figure 2—figure supplement 2A*). The enzymes expressed by the transgene showed high enzymatic activity at Bf, Ft, and Af. At rest time (Rt), the saliva collected from either parotid or mouth showed reduced enzymatic activities. Of note, a significantly lower enzyme activity was observed in the saliva samples collected at Rt (*Figure 2—figure supplement 2B*).

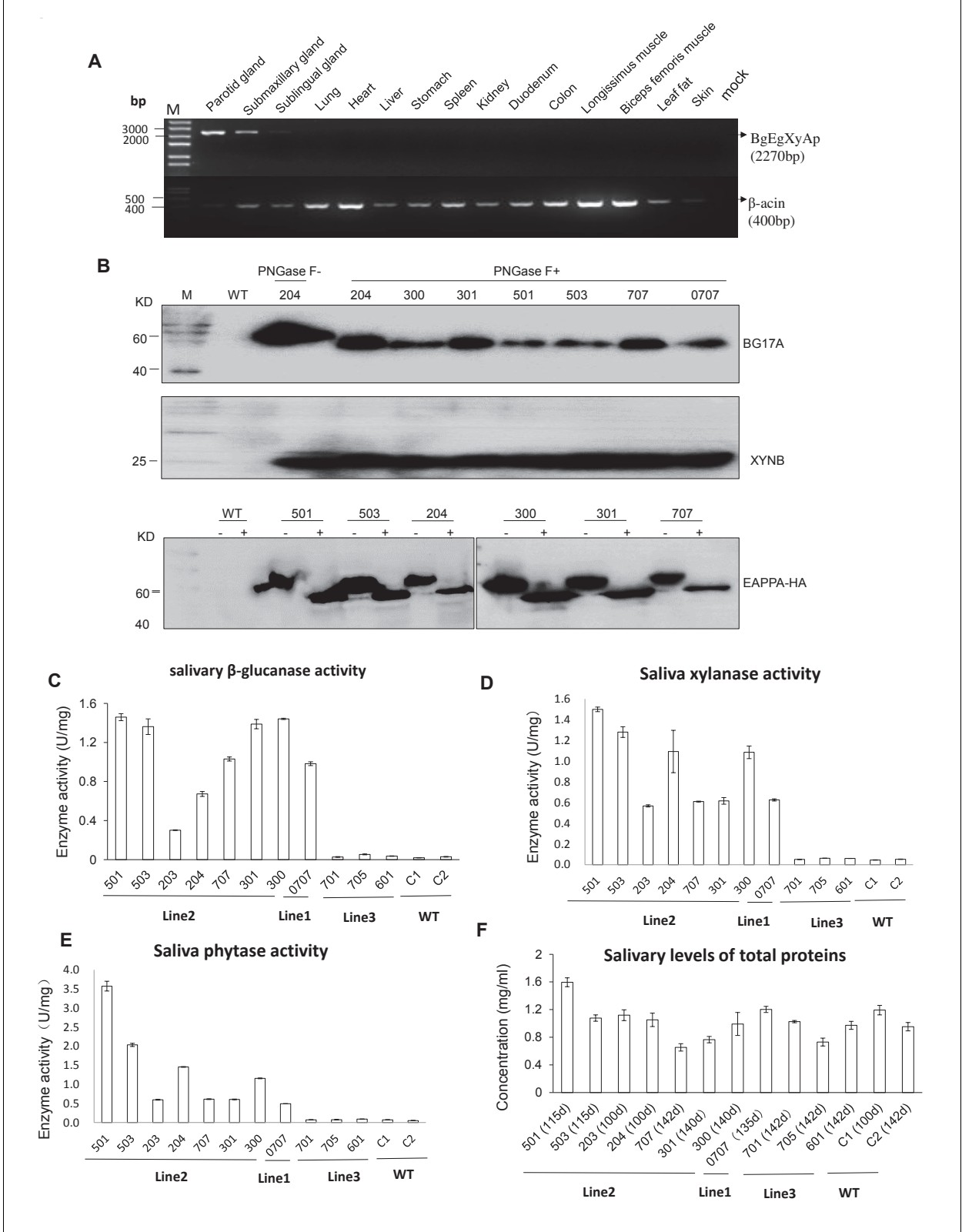

**Figure 2.** Expression and enzymatic activity of transgenes in TG pigs. (**A**) RT-PCR assay for mRNA expression profiles of transgenes in different tissues. Forward primers and reverse primer are bound to the *bg17A* gene and *eappA* gene, respectively (arrows are shown in *Figure 1A*). Mock: Blank control (ddH2O). (**B**) Western blotting assay demonstrating the expression of BG17A, XYNB, and EAPPA in the saliva of TG pigs. Saliva samples were either incubated with PNGase F (+) or mock (-)-treated prior to western blotting to analyze the glycosylation status of the transgenic enzymes. PNGase F: *Figure 2 continued on next page*

*Figure 2 continued*

Peptide N-glycosidase F. (**C–E**) Salivary β-glucanase, xylanase, and phytase activity assays of the TG pigs. (**F**) Concentration of total salivary protein of the TG and WT pigs. C1, C2: Age- and body weight-matched WT pigs. The data presented in the figure (C-E) can be found in *Figure 2—source datas 1–4*.

DOI: https://doi.org/10.7554/eLife.34286.009

The following source data and figure supplements are available for figure 2:

**Source data 1.** Salivary β-glucanase activity assays of TG pigs.

DOI: https://doi.org/10.7554/eLife.34286.012

**Source data 2.** Salivary xylanase activity assays of TG pigs.

DOI: https://doi.org/10.7554/eLife.34286.013

**Source data 3.** Salivary phytase activity assays of TG pigs.

DOI: https://doi.org/10.7554/eLife.34286.014

**Source data 4.** Concentration of total salivary protein of TG and WT pigs.

DOI: https://doi.org/10.7554/eLife.34286.015

**Figure supplement 1.** Assay for mRNA expression levels of the linked transgenes (*BgEgXyAp*) in different organs and tissues of transgenic founder pigs.

DOI: https://doi.org/10.7554/eLife.34286.010

**Figure supplement 2.** Characterization of transgenic (TG) (Line2) enzymes secreted in saliva.

DOI: https://doi.org/10.7554/eLife.34286.011

## Improved feed utilization and reduced nutrient emission in TG founders

The grower-finisher founder TG pigs (weight range: 30 kg to 50 kg) were fed corn-soybean (CS) or wheat-corn-soybean-bran (WCSB) diets (*Supplementary file 2*) to investigate the effects of a salivary cocktail of β-glucanase, xylanase, and phytase on feed utilization. The traditional CS diet contains a low level of NSPs and total P with a high proportion of phytates (70.3%), and the WCSB diet contains a relatively high concentration of NSPs and total P with 63.2% phytates. For each diet, 6 TG pigs and six age-matched and weight-matched non-TG Duroc boars (WTs) were fed; and 6 WTs were fed the same diet with supplementary multi-enzyme preparations of β-glucanase, xylanase, and phytase (namely, WT(+) group). After dietary treatment, nutrient digestion among the experimental groups was measured and compared. For both diets, the apparent total tract digestibility (ATTD) of dry matter (DM), P, N, and calcium (Ca) significantly increased in TG pigs compared with that of WT pigs (*Figure 3A*). The fecal outputs of N and P, relative to input by feed, were significantly decreased in the TG pigs compared with that of the WT pigs (*Figure 3B*). Fecal N and P excretion were decreased by 24.9% and 45.8%, respectively, with the CS diet, and 23.2% and 34.8%, respectively, with the WCSB diet. A significant reduction in total P and Ca (feces plus urine) was also observed in the TG pigs with both diets. Almost all tested parameters of the TG pigs showed some improvement compared with that of the WT(+) pigs that were fed the same diets supplemented with multi-enzyme preparations, although the differences were not statistically significant among groups (*Supplementary file 3* and *Supplementary file 4*).

The serum components of the TG pigs fed on a diet in *Supplementary file 5* containing a low N level and high proportion of phytates (78.4%) (LNHP) were analyzed. Serum alkaline phosphatase activity in the TG pigs was lower than the WT littermates. Serum P and glucose levels of the TG pigs were greater than that of the WT littermates. The serum D-xylose levels of the TG pigs were slightly higher than the WT littermates. No differences in serum Ca, Zn, urea N, uric acid, and total protein concentrations between TG and WT pigs were observed (*Table 2*).

## Enhanced growth performance in TG pigs

To assess the growth performance, eight F1 TG pigs (females) and 17 WT littermates (females) were fed a LNHP diet (*Supplementary file 5*) during the growing period from 30 kg to 50 kg weights was measured. The TG pigs exhibited a higher average daily gain (ADG) rate and lower feed conversion rate (FCR) than the WT pigs during this stage (*Supplementary file 6*). Growth performance of the F2 TG pigs was also measured. A total of 74 F2 TG pigs (23 boars, 51 gilts) and 52 WT littermates (21 boars, 31 gilts) were raised together and fed the same diets shown in *Supplementary file 7*. In this study, TG boars showed increased average daily feed intake (ADFI) (p=0.077) compared to the WT boars that were fed the same diets (*Figure 4A*). Similar results were observed in the gilts.

**Table 1.** Salivary secretion and the transgene enzyme activities from the unilateral parotid gland of transgenic (TG) pigs (Line2) during the grower (92–96 days old; estimated body weight: 42–45 kg) and finisher (159–191 days old; estimated body weight: 100–115 kg) phases of growth.

| | Growth stages | | Pooled SEM | P values |
|---|---|---|---|---|
| Item | Grower | Finisher | | |
| Saliva secreted [*] Saliva secretion rate (mL/min·pig) | 8.48 | 16.97[**] | 0.25 | <0.0001 |
| Saliva secretion volume (mL/kg diet consumed) | 407.66[**] | 151.70 | 5.33 | <0.0001 |
| Enzymes activity secreted (U/mL saliva)[†] | | | | |
| β-glucanase | 5.72 | 6.07[**] | 0.05 | <0.0001 |
| Xylanase | 5.92 | 6.19[**] | 0.05 | 0.0005 |
| Phytase | 7.2 | 6.87 | 0.14 | 0.1033 |
| Enzyme activity secreted (U/kg diet consumed)[†] | | | | |
| β-glucanase | 2,331.84[**] | 920.82 | 30.7 | <0.0001 |
| Xylanase | 2,413.38[**] | 939.03 | 31.73 | <0.0001 |
| Phytase | 2,935.19[**] | 1,042.19 | 38.29 | <0.0001 |

*Saliva samples were collected daily from two transgenic growing pigs and two transgenic finishing pigs at 9:00 and 16:00, respectively, over the four-day period. Values are expressed as the mean and pooled SEM (n = 16 repeated sampling and measurements).

†Means and pooled SEM (n = 16) repeated sampling and measurements). Asterisks indicate significant differences between the grower and the finisher phases within the same row (Unpaired t-test, **$p < 0.01$).

DOI: https://doi.org/10.7554/eLife.34286.016

The following source data available for Table 1:

**Source data 1.** Salivary secretion from the unilateral parotid gland in the growing transgenic (TG) pigs and wild type (WT) pigs during the grower phase (92–96 days old; estimated body weight of 42–45 kg).
DOI: https://doi.org/10.7554/eLife.34286.017

**Source data 2.** Salivary secretion from the unilateral parotid gland in the transgenic (TG) pigs during the finisher phase (159–191 days old; estimated body weight of 100–115 kg).
DOI: https://doi.org/10.7554/eLife.34286.018

**Source data 3.** The transgene enzyme activities from the unilateral parotid gland in the transgenic (TG) pigs during the grower (92–96 days old; estimated body weight of 42–45 kg) and the finisher phases (159–191 days old; estimated body weight of 100–115 kg).
DOI: https://doi.org/10.7554/eLife.34286.019

Significantly improved ADG rates and lower FCR were observed in the TG boars and gilts compared to the WT gilts during the entire feeding period (*Figure 4B and C*). It took an average of 110 days for TG boars to grow from 30 kg to 115 kg, whereas the WT boars needed 145 days of feeding on the same diets. Similar results were observed in the gilts. It took 121 days for the TG gilts to grow from 30 kg to 115 kg, whereas the WT females required 150 days (*Figure 4D*). Taken together, the TG pigs showed a 7.0–7.6% higher ADFI than the WT pigs over the same grower and finisher phases (30 kg to 115 kg), but gained 23.0–24.4% more body weight (BW) daily than the WT pigs over the same grower and finisher phases (30 kg to 115 kg). The time to reach 115 kg BW was shortened by 19.2–21.9% (29.6 days to 35.1 days). The FCR decreased by 11.5–14.5% during the grower and finisher periods.

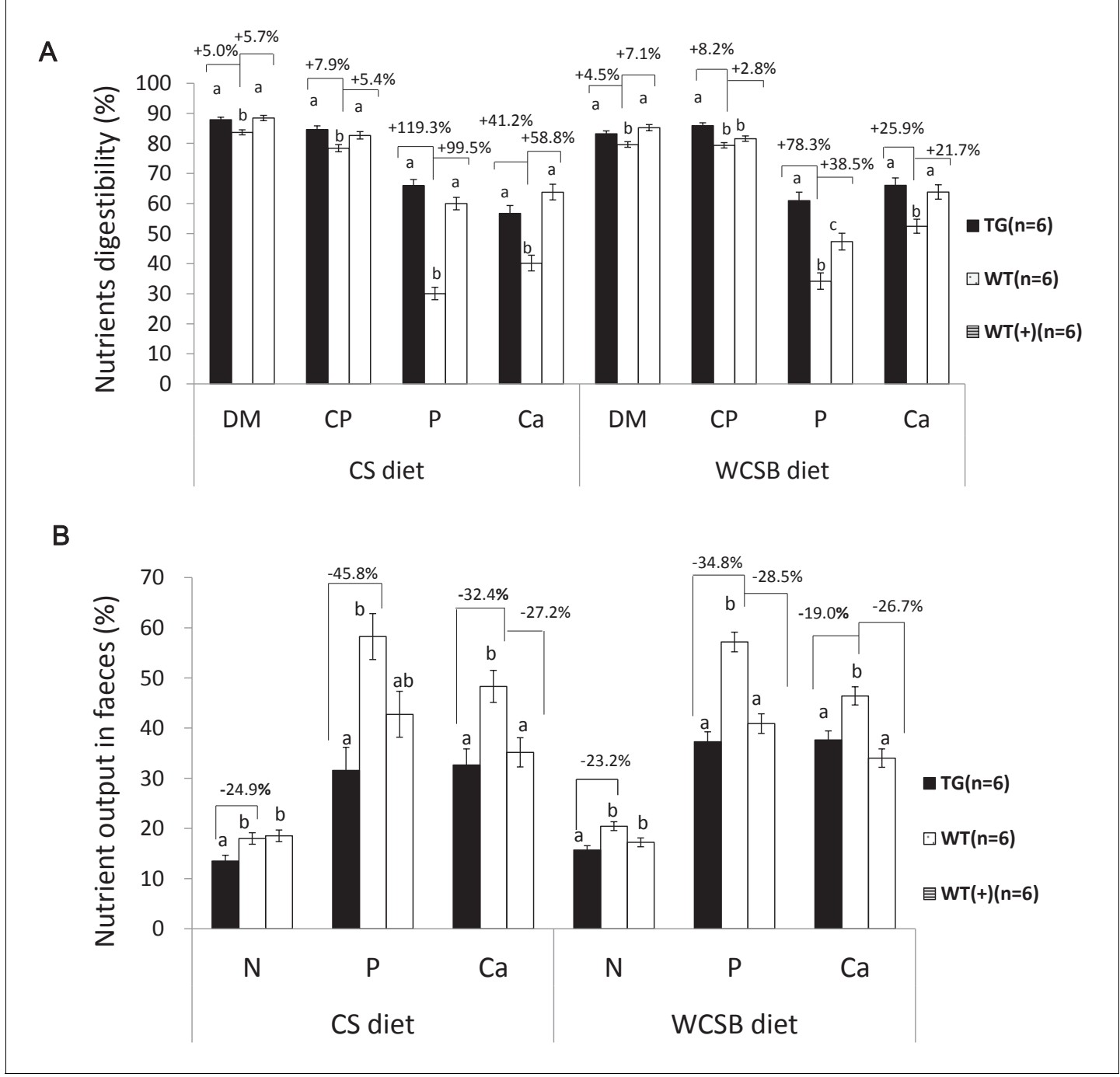

**Figure 3.** Comparison of the apparent total tract nutrient digestibility values (%) and fecal nutrient output (% of their dietary intake) between transgenic (TG) grower pigs (Line2) and their wild-type (WT) littermates fed on corn and soybean meal (CS) and wheat- and corn and soybean meal (WCSB)-based diets with and without exogenous feed enzymes. (**A**) Comparison of the apparent total tract nutrient digestibility values (%) of dry matter (DM), crude protein (CP), phosphorus (P), and calcium (ca). (**B**) Comparison of fecal N, P, and Ca output. WT(+): WT grower pigs fed on the CS and WCSB diets supplemented with an optimal dose of β-glucanase, xylanase, and phytase. Data are expressed as the least square means (Lsmean ± SEM). [a,b,c] Values on the bar graph with different superscript letters differ significantly (ANCOVA, p<0.05). The source data are presented in *Figure 3—source datas 1–6*.
DOI: https://doi.org/10.7554/eLife.34286.020

The following source data is available for figure 3:

**Source data 1.** Comparison of the apparent total tract nutrient digestibility values (%) of dry matter(DM), crude protein(CP), Phosphorus(P) and calcium(ca).
DOI: https://doi.org/10.7554/eLife.34286.021

*Figure 3 continued on next page*

*Figure 3 continued*

**Source data 2.** Comparison of the apparent total tract nutrient digestibility values (%) between transgenic (TG) grower pigs and their wild-type (WT) littermates fed corn and soybean meal (CS diet).
DOI: https://doi.org/10.7554/eLife.34286.022

**Source data 3.** Comparison of the apparent total tract nutrient digestibility values (%) between transgenic (TG) grower pigs and their wild-type (WT) littermates fed wheat, corn and soybean meal (WCSB diet).
DOI: https://doi.org/10.7554/eLife.34286.023

**Source data 4.** Comparison of fecal N, fecal P and fecal Ca output.
DOI: https://doi.org/10.7554/eLife.34286.024

**Source data 5.** Source Data of fecal N, fecal P and fecal Ca output by transgenic (TG) grower pigs and their wild-type (WT) littermates fed corn and soybean meal (CS)based diets with and without exogenous feed enzymes.
DOI: https://doi.org/10.7554/eLife.34286.025

**Source data 6.** Source Data of fecal N, fecal P and fecal Ca output by transgenic (TG) grower pigs and their wild-type (WT) littermates fed wheat, corn and soybean meal (WCSB) based diets with and without exogenous feed enzymes.
DOI: https://doi.org/10.7554/eLife.34286.026

## Discussion

Previous studies have shown that TG pigs that secrete the microbial enzyme phytase from their salivary glands have significantly reduced P levels in their manure (*Golovan et al., 2001*; *Meidinger et al., 2013*). The major objective of the present study was to enhance the digestive utilization of feed grain and decrease the N and P emissions from pig manure. We obtained TG pigs by introducing the microbial genes *bg17A*, *eg1314*, *xynB*, and *eappA*, which encode for endo-glucanase (the glycoside hydrolase family 7), endo-β−1,3–1,4-glucanase (the glycoside hydrolase family 16), endo-xylanase, and 6-phytase, respectively, into the genome of pigs so that these could produce phytate- and NSP-degrading enzymes. These enzymes, which are inherently secreted by microbial communities, were optimized to adapt to the digestive tract environment of pigs and expressed specifically in the porcine salivary gland to initiate the digestion process of NSPs and phytates in the mouth. The feeding trials demonstrated that the TG pigs possessed a high digestive capacity for N, DM, phytate P, and other nutrients, enhanced growth performance, and decreased manure nutrient

**Table 2.** Comparison of the serum biochemical endpoints in the F1 transgenic (TG) grower gilts (Line2) and the wild-type (WT) gilts (50 kg) fed on the LNHP diet in *Supplementary file 5*.

| Serum component | TG (n = 10) | WT (n = 22) | Pooled SEM | P value |
| --- | --- | --- | --- | --- |
| Alkaline phosphatase (U/L) | 84.90** | 155.72 | 12.38 | <0.01 |
| Total phosphorus (Pi) (mmol/L) | 3.05** | 2.09 | 0.13 | <0.01 |
| Total calcium (Ca) (mmol/L) | 2.53 | 2.48 | 5.51 | 0.52 |
| Urea N (mmol/L) | 3.27 | 3.78 | 0.39 | 0.39 |
| Glucose (mmol/L) | 4.97** | 3.78 | 0.21 | <0.01 |
| Uric acid (μmol/L) | 6.30 | 7.45 | 0.54 | 0.13 |
| D-Xylose (mmol/L) | 0.39 | 0.35 | 0.01 | 0.07 |
| Total protein (g/L) | 59.87 | 56.55 | 2.16 | 0.35 |
| Zn (μmol/L) | 4.12 | 5.23 | 0.82 | 0.42 |

Samples were collected at the end of the experiment. Asterisks (**) indicate significant differences at $p < 0.01$ (unpaired t-test) between TG and WT pigs within the same row.
DOI: https://doi.org/10.7554/eLife.34286.027
The following source data available for Table 2:
**Source data 1** Comparison of the serum biochemical composition in the F1 transgenic (TG) grower gilts and the wild-type (WT) gilts (50 kg) fed the low non-starch polysaccharide (NSP) diet.
DOI: https://doi.org/10.7554/eLife.34286.028

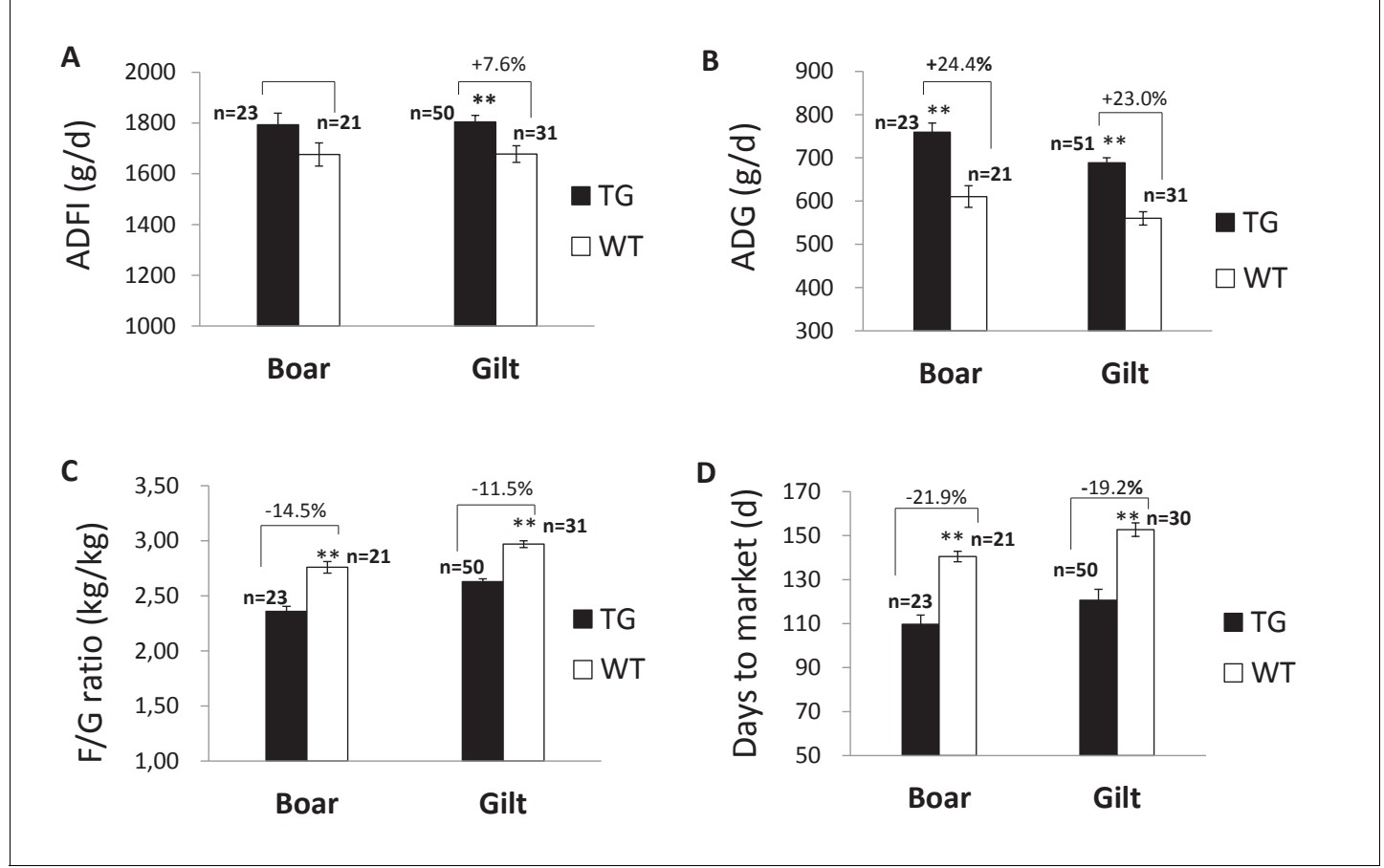

**Figure 4.** Growth performance of F2 TG pigs (line2) and WT littermates during the growing period from 30 kg to 115 kg. (**A**) Comparison of average daily feed intake (ADFI). (**B**) Comparison of average daily gain (ADG). (**C**) Comparison of feed/gain (F/G). (**D**) Comparison of days to market. Data are expressed as the least square means (Lsmean ± SEM), asterisks indicate significant differences between TG and WT pigs within one line (ANCOVA, **p<0.01). The source data are presented in *Figure 4—source datas 1–3*.

DOI: https://doi.org/10.7554/eLife.34286.029
The following source data is available for figure 4:

**Source data 1.** Growth performance of F2 TG pigs (boars + gilts)and WT littermates during the growing period from 30 kg to 115 kg.
DOI: https://doi.org/10.7554/eLife.34286.030
**Source data 2.** Growth performance of F2 TG and WT boars during the growing period from 30 kg to 115 kg.
DOI: https://doi.org/10.7554/eLife.34286.031
**Source data 3.** Growth performance of F2 TG and WT gilts during the growing period from 30 kg to 115 kg.
DOI: https://doi.org/10.7554/eLife.34286.032

emissions. We did not find any negative side effects in these TG animals such as changes in spirit, behavior, reproductive capacity, viability, growth performance, blood physiology, and biochemistry.

We totally obtained 33 live cloned piglets. Twenty-five of them were positive for the transgene. Although we used the selected single-cell-colonies for SCNT, these colonies might include few cells from other colonies with different genotypes as all colonies were cultured and screened on the same dish. Therefore, the transgene-negative pigs will be produced if impure cell colonies were used for animal cloning. Many of the piglets born alive died from visible malformation (30.3%) and weakness (18.2%) likely related to various internal malformation in the digestive, circulatory or cardiopulmonary systems. A previous study reported that WT and TG cloned pigs showed 10.9–17.8% and 25.2–39.8% malformation rates, respectively, which resulted in a considerable loss (58%) of cloned piglets before weaning (*Liu et al., 2015a*; *Schmidt et al., 2015*). In our result, the malformation and mortality rate was in agreement with the previous report. Notably, the non-TG cloned pigs in our study also showed a high death rate, suggesting that the problem of high mortality rate of transgenic

cloned pigs may relate to cloning technique. The insertion sites of transgene are the intron region of *Legumain* and *CEP112* for line 1 and line 2, respectively. Legumain is a cysteine protease that involves in antigen processing for class II MHC presentation in lysosomes (*Chen et al., 1997*). Legumain is overexpressed in some solid tumors (*Dall and Brandstetter, 2016*). Previous studies reported that inhibition/knock-down of *legumain* could suppress tumor progression (*Briggs et al., 2010*; *Liu et al., 2015b*). Legumain knockout mice showed a phenotype of decreased weight gain (*Shirahama-Noda et al., 2003*). *CEP112* gene encodes a centrosomal protein of 112 kDa which is localized around spindle poles, and its structure and function are still unclear (*Jakobsen et al., 2011*; *Kumar et al., 2013*). The expression and function of the two genes in TG pigs may not be compromised significantly, as the foreign transgene is integrated into the intron region. No significant phenotypic difference was observed between the TG pigs and their WT littermates.

In this study, transgenes encoding microbial enzymes were driven by the salivary gland-specific PSP promoter. TG pig line two harbors a single copy of a multi-transgene, which facilitates the establishment of a breeding population with identical genotypes. Compared to the previously reported phytase TG pig lines that harbor a high copy number of a foreign gene in an insertion locus (2 to 35 copies of the transgene, producing 5 to 6,000 U/mL of phytase) (*Golovan et al., 2001*), our multi-TG pigs secreted a lower amount of enzymes into the saliva. In fact, our multi-TG pigs demonstrated similar digestibility with the high-expression phytase TG pigs, as well as reduced emission of P in manure. For grower-finisher pigs, our study displayed an approximately 45.7% reduction in fecal P when fed CS meals (70.3% phytates), versus 56% reportedly in high-expression phytase TG pigs fed on soybean meals (53% phytates) (*Golovan et al., 2001*). The three major salivary glands showed high expression levels and enzymes activities, which could be arranged in decreasing order as follows: parotid gland >sublingual gland>submaxillary glands, which agree with the findings of a previous report (*Golovan et al., 2001*). The enzyme activities per milliliter of parotid saliva between grower pigs and finisher pigs were similar. The expression levels of glucanase, xylanase, and phytase in the bilateral parotid glands of the TG pigs were 4,663.68, 4,826.76, and 5,870.38 U/kg diet intake for grower pigs, and 1,841.64, 1,878.06, and 2,084.38 U/kg diet intake for finisher pigs. These levels of expression are higher than the amounts used as dietary supplements.

The TG pigs demonstrated increased P retention and reduced percentages of manure P excretions compared with age-matched WT pigs fed on the same diets, implying the significant digestive effect of TG phytase. The P retention rate was 54.7–67.3%, and reduction in total P output ranged from 21.3% to 44.2%. These results are slightly lower than that of previously reported TG phytase pigs, which had P retention rates of 57.2–77.8%, and total P output decreased by 27.5–62.0% (*Golovan et al., 2001*; *Meidinger et al., 2013*). This discrepancy may be caused by the different expression levels of transgenes and the ingredients of the diets. However, the new pig model demonstrated that N digestibility was significantly enhanced in all diets tested. WCSB diets contain high levels of NSPs, which account for the highest fraction of polymeric carbohydrate in protein-rich materials (aleurone layer). Glucanase and xylanase secreted by TG pigs can effectively degrade the glucans and xylans in the cell walls of the aleurone layer, in which the matrix proteins and phytate globoids would be exposed and subjected to degradation by phytase and endogenous proteases. These results agree with that of high-NSP diets supplemented with NSPase (*Diebold et al., 2004*; *Willamil et al., 2012*). For the corn-soybean diet, the effects of enzyme supplementation on nutrient digestibility of pigs, as reported in the literature, were highly variable. Some studies show positive responses to enzyme supplementation (*Ji et al., 2008*; *Kim et al., 2003*), whereas others do not (*Li et al., 1999*; *Willamil et al., 2012*). TG pigs fed on CS diets also showed significantly enhanced N digestibility. As the main ingredient of CS diets, corn contains high levels of phytates. The synergistic effect of NSPase (glucanase and xylanase) and phytase involves the digest of NSPs and phytates in the aleurone layer and endosperm cells of corn, thereby alleviating the anti-nutrition effect of phytates by reducing the binding of phytates to proteins and digestive enzymes in the digestive tract. In terms of N emission, the grower TG pigs emitted 24.0% less fecal N than the WT pigs. The NSPs in the cell wall are degraded by NSPases (glucanase and xylanase) that are secreted in the upper digestive tract. Moreover, NSPases can reduce the compensated secretion of endogenous fluids to decrease endogenous nutrient losses (*Kerr and Shurson, 2013*). TG phytase facilitates in the release of phytate-chelating proteases and other digestive enzymes to further degrade nutrients.

The TG pigs displayed better growth performance than conventional pigs fed on diets without supplemental P during the grower and finisher phases. Both TG boars and gilts showed a faster rate

in BW gain, and exhibited greater feed efficiency than the respective WT littermates. Blood serum measurements also demonstrated enhanced serum P, glucose, and xylose levels in TG pigs, which signified enhanced digestive utilization of these nutrients. A significant decrease in serum alkaline phosphatase levels is an indicator of well-developed bone (*Meidinger et al., 2013*; *Sefer et al., 2012*; *Selle and Ravindran, 2008*). These results are clear manifestations of the improved nutrient digestibility and growth performance of TG pigs.

In summary, the multi-TG pigs reported in the present study exhibited significantly enhanced digestibility of feed N, P, Ca, and other minerals. The pigs produced manure with significantly reduced P and N emissions into the environment as well as exhibited improved growth performances. This genetic strategy offers a very valuable biological solution for inefficient feed digestion and environmental emissions due to the global expansion of the livestock industry.

## Materials and methods

### Transgene constructs

The codons of two β-glucanase genes, *bgl7A* from *Bispora* sp. MEY-1 (*Luo et al., 2010*) and *eg1314* from *Bacillus licheniformis* EGW039 (CGMCC 0635) (*Teng et al., 2006*), three xylanase genes, *xyl11*, *xyn63*, and *xynB*, from *Aspergillus niger* CBS513.88 (*Liu et al., 2010*), *Penicillium* sp. F63 CGMCC1669 (*Deng et al., 2006*), and *Aspergillus niger* CGMCC1067 (*Guo et al., 2013*), (respectively), as well as two phytase genes, *eappA* (GenBank accession No. AF537219.1) from *Escherichia coli* and *cappA* (GenBank accession No. AF537219.1) from *Citrobacter freundii* were optimized per codon usage bias in pigs. The original signal peptide of these genes were replaced by the signal peptide of the porcine parotid secretory protein (PSP). Optimized genes were synthesized by Genscript (Nanjing, China) and ligated into the eukaryotic expression vectors pCDNA3.1+or pcDNA6.0 (Invitrogen, Carlsbad, CA, USA).

The *bgl7A*, *eg1314*, *xynB*, and e*appA* genes were fused in a head-to-tail tandem array, with E2A, T2A, and P2A used as linkers between them. Flag-tag and HA-tag were added to the C terminal of Eg1314 and EAPPA, respectively, to facilitate the detection of protein expression. The fusion of the four genes, named *BgEgXyAp*, was cloned into pCDNA3.1(+) to examine its expression and enzyme activity levels by transient transfection using Lipofectamine 2000 (Invitrogen Carlsbad, CA, USA) in porcine cells. A 12.1 kb upstream genomic sequence of murine PSP, as a promoter to drive the expression of the fusion gene specifically in salivary gland, were cloned and ligated to *BgEgXyAp*, and then introduced into the transposon piggyBac vector pPB-lox-*neoEGFP*-loxp (a gift from The Wellcome Trust Sanger Institute, Cambridgeshire, UK) to form the final transgene construct. The final construct was confirmed by sequence analysis.

### Transgenic pigs

Primary PFFs were isolated from 35-day-old male fetuses of Duroc pigs. PFFs were cultured in Dulbecco's Modified Eagle Medium (DMEM, Gibco 12491–023) (Thermo Fisher Scientific, Suwanee, GA, USA) supplemented with 12% fetal bovine serum (FBS, Gibco 10100–147) (Thermo Fisher Scientific, Suwanee, GA, USA) and 1% (v:v) penicillin/streptomycin (10,000 U/mL penicillin, 10,000 μg/mL streptomycin; GIBCO-BRL, Grand Island, NY, USA) at 39°C in an incubator with 5% $CO_2$. The transgene was mixed with a transposase, pCMV-hyPBase (a gift from the University of Hawaii, Honolulu, HI), and transfected into PFFs by electroporation (BTX, San Diego, CA). The transfected cells were split 1:6 into fresh culture medium. After 24 hr, 300 μg/mL G418 (Gibco) was added to the medium to select transfected cell colonies, and the plates were incubated in media containing G418 for about 15 days. The surviving cell colonies with EGFP expression were isolated within colony cylinders (Bellco Glass, Vineland, NJ, USA), and propagated in a fresh 24-well plate. Four colonies that proliferated well, with bright fluorescence, were then expanded and screened for the presence of the *BgEgXyAp* transgene.

SCNT was performed as previously described (*Lai et al., 2006*). The reconstructed embryos were surgically transferred to the oviduct of the recipient gilts the day after estrus was observed. The pregnancy status of the surrogates was detected using an ultrasound scanner at 26 d after the embryo was transferred. This status was then monitored weekly before the expected due date. The cloned piglets were born by natural birth.

## PCR and quantitative PCR

Genomic DNA was isolated from pig ear skin biopsies and used in PCR identification of the transgene. The PCR primes are listed in *Supplementary file 8*. For RT-PCR, reverse transcription of mRNA was conducted to generate cDNA, which was then used as template for PCR. The RT-PCR primers are listed in *Supplementary file 9*. Relative quantitative real-time PCR and absolute quantitative real-time PCR used in detecting expression levels and copy numbers of transgene, respectively, were performed as described elsewhere (*Wu et al., 2013*). The Q-PCR primers are listed in *Supplementary file 9*.

## Southern blotting

Genomic DNA was isolated from the ears of transgenic founders and wild-type (WT) controls by phenol-chloroform extraction. Fifteen micrograms of DNA were digested with *Hind*III, fractionated in a 0.8% agarose gel, and transferred onto a nylon membrane (GE Healthcare, Pittsburgh, PA, USA). The membrane was then hybridized with a probe. The probe primers are listed in *Supplementary file 8*. Hybridization and washing were performed with DIG-High Prime DNA Labeling and Detection Starter Kit II (Roche, Basel, Switzerland). Prehybridization was conducted at 42°C for 30 min, hybridization at 50°C for 8 hr, then incubated for 30 min in blocking solution and further incubated for 30 min in an anti-digoxigenin-AP antibody solution. After incubation, the membrane was exposed for 5–20 min to 1 mL of ready-to-use CSPD, and images were captured with an EC3 imaging system (UVP, LLC, Upland, CA, USA).

## Western blotting

Pig saliva was concentrated using an Amicon Ultra 15 mL centrifugal filter (Millipore). After protein quantification, total protein samples were separated on a 10% sodium dodecyl sulfate polyacrylamide gel (SDS-PAGE), and transferred to a polyvinylidene difluoride (PVDF)(Millipore, Temecula, CA, USA) membrane. The membranes were blocked with 5% non-fat dry milk, subsequently incubated with the corresponding antibodies, and then developed with an enhanced chemiluminescence solution (Thermo Fisher Scientific, Suwanee, GA, USA). Chemiluminescent signals were captured by a cooled charged-coupled device (CCD) camera. For the primary antibodies, rabbit polyclonal anti-BG17A and anti-XYNB antibodies, which were prepared by Genscript (Nanjing, China), were respectively used to detect β-glucanases and xylanase. A mouse monoclonal anti-HA antibody (Abcam, Cambridge, UK) was used to detect phytase. Information on the antibodies used in this study is presented in *Supplementary file 10*. Horseradish peroxidase-conjugated anti-mouse IgG or horseradish peroxidase-conjugated anti-rabbit IgG (Abcam, Cambridge, UK) was used as secondary antibody.

## Saliva collection

Oral saliva was collected by clamping an absorbent cotton into the buccal cavity, and allowing the piglets to soak with saliva and chew. The resulting liquid was squeezed out using an injector, centrifuged, and then used for assays. To collect saliva from the parotid gland, a fistula was surgically installed under anesthesia with propofol at the unilateral (right cheek) parotid duct of each pig. Saliva from the parotid gland was collected using a medical drainage bag during feeding time, before feeding (within 30 min), after feeding (within 30 min) and at selected time points dur

## Enzymatic activity assay

The supernatants from transfected cells and saliva from transgenic and non-transgenic pigs were used as total protein samples for the enzymatic activity assays. β-glucanase and xylanase activity assays were based on estimating the amount of reducing sugars released from the relevant substrates in the reactions using 3,5-dinitrosalicylic acid (DNS) reagent, as previously described (*Liu et al., 2010*; *Luo et al., 2010*). One unit of activity was defined as the quantity of enzyme that releases reducing sugar at the rate of 1 μmol/min.

Phytase activity in saliva was determined by means of vanadium molybdenum yellow spectrophotometry. The reaction was performed in a final volume of 600 μL solution containing 0.25 M of acetate buffer (pH 5.5), 5 mM sodium phytate, and 50 μL enzyme preparation at 39°C for 30 min, followed by termination of reaction by adding 400 mL of an ammonium molybdate-ammonium vanadate-nitric acid mixture. After mixing and centrifugation, the absorbance was measured at a

wavelength of 415 nm. One unit of phytase activity was defined as the amount of activity that liberates one micromole of phosphate per minute at 39°C.

### Dietary treatments

Six TG pigs and 12 age- and body weight-matched non-TG pigs were used in dietary treatment experiments. The TG group and six non-TG pigs were fed a diet (CS or WCSB). The other six non-TG pigs were fed the same diet with exogenous feed enzymes. The pigs were housed in small groups of three to four animals per pen. Each pig was kept in individual metabolic cage (Length × width × height: 1.40 × 0.67 × 1.15 m) with stainless steel mesh floors for collection of urine and feces sample at the indicated time points. The cage was laid side by side for pig to communicate with each other. The facilities were provided with forced ventilation and heat lamp for thermal regulation, and each cage had one feeder and one water nipple for *ad libitum* access to feed and water. The ingredients of the selected experimental diets are listed in *Supplementary file 2*. The diets were provided in pellet form.

All pigs were fed on a commercial grower diet for an adaptation period of one week in the cage, and then fed experimental diets for a pre-test period of six days prior to the start of the experiment. During the dietary treatment period, the animals were fed the experimental diets for four days. Pig feed intake was based on BW (BW ×0.04). Equal quantities of diets were added to the feeders twice daily in the morning (09:00) and afternoon (16:00), and the initial and final BW of each experimental diet were recorded.

Fecal samples were collected and processed as previously described (*Kim et al., 2005*). Feeds and dried feces were ground using a grinder, passed through a 0.425 mm size screen, and analyzed for DM (AOAC, 2005; method 930.15). Gross energy (GE) was analyzed according to ISO: 9831–1998 using a bomb calorimetry (Parr 6300; Parr Instrument Co., Louisville, KY, USA). Other nutrients in the diets and feces were analyzed using the Chinese National Standard analytical method (GB/T). The following methods were used: GB/T 6432–1994 for CP, GB/T 20806–2006 for NDF, GB/T 20805–2006 for ADF, GB/T 6434–2006 for CF, GB/T 18335–2003 for calcium, GB/T 6437–2002 for P, GB/T 6438–2007 for ash, and GB/T 21912–2008 for titanium dioxide. Apparent digestibility coefficients were calculated as described elsewhere (*Grela et al., 20112018*).

### Growth Performance assessment

The growth of the F2 TG pigs of both genders was compared to that of age-matched WT littermates fed on the same diets listed in *Supplementary file 7*. A total of 23 TG boars (32.9 ± 3.80 kg) and 21 WT littermates boars (31.5 ± 2.97 kg) were grouped according to weight, and randomly allocated to seven pens fitted with MK3 FIRE feeders (**FIRE**, Osborne Industries Inc., Osborne, KS, USA). Similarly, 51 TG gilts (31.8 ± 3.55) and 31 WT gilts (30.2 ± 1.64) were randomly allocated to 11 pens that were fitted with MK3 FIRE feeders. Individual feed intake and BW were recorded when a pig with an ear transponder visited the FIRE feeders. All pigs were allowed free access to water throughout the measurement phase.

### Statistical analyses

The data were analyzed using the GLM procedure (SAS Inst. Inc., Cary, NC, USA). For the apparent total tract nutrient digestibility values, fecal nutrient output, and the growth performance, analysis of covariance (ANCOVA) was used. The BW of the tested pigs at the start of the corresponding experimental period was used as the covariable. Least square means were calculated and the differences between means were tested using Turkey-Kramer adjustment for multiple comparisons when appropriate. For salivary protein and saliva secretion by the parotid gland, one-way ANOVA followed by Duncan's multiple-comparison tests were used. For serum biochemical endpoints and saliva enzymes secretion, an unpaired two-sample *t*-test (two-tailed) was used. The level of significance was set at $p < 0.05$, and trends were discussed at $p < 0.1$.

## Acknowledgements

We thank Dr. Hao Zhang (South China Agricultural University, Guangzhou, China) for data analysis and to Dr. Stefan Moisyadi (University of Hawaii, Honolulu, HI, USA) for providing transposon

plasmids. We thank Dr. Defu Zhang and Dr. Yufang Ma for providing the *cappA* gene. National Science and Technology Major Project for Transgenic Breeding (2016Z × 008006002) supported this study.

## Additional information

### Funding

| Funder | Grant reference number | Author |
|---|---|---|
| National Science and Technology Major Project for Transgenic Breeding | 2016ZX008006002 | Zhenfang Wu |

The funders had no role in study design, data collection and interpretation, or the decision to submit the work for publication.

### Author contributions

Xianwei Zhang, Data curation, Formal analysis, Investigation, Methodology, Writing—original draft, Writing—review and editing; Zicong Li, Resources, Formal analysis, Investigation, Project administration; Huaqiang Yang, Formal analysis, Methodology, Writing—original draft, Writing—review and editing; Dewu Liu, Conceptualization, Resources, Supervision, Project administration; Gengyuan Cai, Conceptualization, Resources, Funding acquisition, Project administration; Guoling Li, Data curation, Software, Investigation, Methodology; Jianxin Mo, Dehua Wang, Cuili Zhong, Yue Sun, Data curation, Investigation, Methodology; Haoqiang Wang, Data curation, Investigation; Junsong Shi, Miaorong Huang, Resources, Investigation, Methodology; Enqin Zheng, Investigation, Project administration; Fanming Meng, Resources, Methodology, Project administration; Mao Zhang, Xiaoyan He, Rong Zhou, Jian Zhang, Ran Zhang, Resources, Investigation; Ning Li, Conceptualization, Resources, Methodology; Mingzhe Fan, Data curation, Formal analysis, Writing—review and editing; Jinzeng Yang, Resources, Methodology, Writing—review and editing; Zhenfang Wu, Conceptualization, Resources, Supervision, Funding acquisition, Validation, Project administration, Writing—review and editing

### Author ORCIDs

Xianwei Zhang  http://orcid.org/0000-0001-6205-3050
Zicong Li  http://orcid.org/0000-0002-6997-4669
Huaqiang Yang  http://orcid.org/0000-0002-4287-0026
Zhenfang Wu  http://orcid.org/0000-0002-5586-6771

### Ethics

Animal experimentation: Animal use followed the Instructive Notions with Respect to Caring for Laboratory Animals, issued by the Ministry of Science and Technology of China, and the NIH Guide for the Care and Use of Laboratory Animals. The animal use protocol was approved by the Institutional Animal Care and Use Committees (IACUCs) of South China Agricultural University (Approval No. 2013-P001). All surgery was performed under anesthesia with isoflurane or propofol, and all effort was made to reduce the number of animals used and to minimize animal suffering.

### Decision letter and Author response

Decision letter https://doi.org/10.7554/eLife.34286.046
Author response https://doi.org/10.7554/eLife.34286.047

## Additional files

### Supplementary files

• Supplementary file 1. Integration site analysis of the TG pigs
DOI: https://doi.org/10.7554/eLife.34286.033

• Supplementary file 2. Ingredients and nutrient composition of corn-cottonseed meal-rapeseed meal-soybean meal-based (CS) and wheat-corn-soybean meal-based (WCSB) diets for examining the efficiency of nutrient utilization in transgenic grower pigs (weight range: 35–54 kg)
DOI: https://doi.org/10.7554/eLife.34286.034

• Supplementary file 3. Comparison of the apparent total tract nutrient digestibility values (%) between transgenic (TG) grower pigs (Line2) and their wild-type (WT) littermates fed on corn and soybean meal (CS)- and wheat, corn, and soybean meal (WCSB)- based diets with and without exogenous feed enzymes.
DOI: https://doi.org/10.7554/eLife.34286.035

• Supplementary file 4. Comparison of efficiency of dietary nitrogen (N), phosphorus (P), and calcium (Ca) retention (% of their dietary intake) between transgenic (TG) grower pigs (Line2) and their wild-type (WT) littermates fed on the CS and corn-soybean (CS) or wheat-corn-soybean-bran (WCSB) diets with and without exogenous feed enzymes.
DOI: https://doi.org/10.7554/eLife.34286.036

• Supplementary file 5. Ingredients and nutrient composition of the low nitrogen level and high proportion of phytate (78.4%) (LNHP) diet used to assess growth performances of the F1 transgenic (TG) grower gilts and the wild-type (WT) grower gilts (weight range: 30–50 kg).
DOI: https://doi.org/10.7554/eLife.34286.037

• Supplementary file 6. Comparison of the growth performances of the F1 transgenic (TG) gilts (Line2) and the wild-type (WT) gilts fed on a low non-starch polysaccharide (NSP) diet during the growing period (weight range: 30–50 kg)
DOI: https://doi.org/10.7554/eLife.34286.038

• Supplementary file 7. Ingredients and nutrient composition of the experimental diets that were formulated to determine the growth performance of the F2 transgenic (TG) grower-finisher pigs
DOI: https://doi.org/10.7554/eLife.34286.039

• Supplementary file 8. Primers used in PCR and probes in southern blotting
DOI: https://doi.org/10.7554/eLife.34286.040

• Supplementary file 9. Primers used in reverse transcription PCR, quantitative real-time PCR, and absolute quantitative real-time PCR
DOI: https://doi.org/10.7554/eLife.34286.041

• Supplementary file 10. Antibody used in western blotting
DOI: https://doi.org/10.7554/eLife.34286.042

• Supplementary file 11. Efficiency of piggyBac-mediated transgenisis to produce single-copy quad-cistronic transgenic pigs
DOI: https://doi.org/10.7554/eLife.34286.043

• Transparent reporting form
DOI: https://doi.org/10.7554/eLife.34286.044

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
