## [Decision Letter]

Thank you for submitting your article "Generation of a novel growth-enhanced and reduced environmental impact transgenic pig strain" for consideration by *eLife*. Your article has been reviewed by three peer reviewers, and the evaluation has been overseen by a Reviewing Editor and Ian Baldwin as the Senior Editor. The following individuals involved in review of your submission have agreed to reveal their identity: Simon Lillico (Reviewer #1); Björn Petersen (Reviewer #3).

The reviewers have discussed the reviews with one another and the Reviewing Editor has drafted this decision to help you prepare a revised submission.

Summary:

The manuscript reports the production and characterization of pigs that were transgenic for several microbial enzymes expressed from the salivary glands. The expression of these enzymes is demonstrated, and the desired biological effects can be observed. Fecal nitrogen and phosphorous outputs were reduced by 20 to 46% , growth rate was improved by about 24% . The feed conversion rate was also improved by 12 to 14% . The authors claim that this is a promising approach to improve feed efficiency and to reduce emissions into the environment. Zhang et al., present a very interesting and important pig model for the expression of multiple microbial transgenes to improve the digestibility of pig feed and to reduce the excretion of N and P into the environment.

Essential revisions:

1) Although single cell clones positive for the transgene and EGFP were used for nuclear transfer, only 24 of the 35 piglets born were transgenic. The authors should discuss possible reasons why this happened.

2) 8 of these pigs survived to sexual maturity. 4 piglets contained the plasmid backbone (ampicillin gene), which the authors claimed was due to breaks in the transgene. This could simply be a consequence of random transgene integration and not transposase mediated integration and the authors should adjust their explanation to reflect this likely possibility. The other pigs contained either a single copy integration in the Legumain or CEP112 gene or a triple integration in an intergenic region. It is unclear whether only four clones were used for nuclear transfer, and if so why these four clones were chosen. The authors should clarify these points.

3) The authors also need to give a brief description of the two genes (Legumain or CEP112) and if their expression was affected by the insertion of the transposon. Breeding was carried out with the line in which the transposon had integrated in CEP112, but no explanation is provided why this line was chosen.

4) In the feeding experiments the authors could show the functionality of the transgenes. Reduction of phosphorous output was however lower than pigs produced in 2001. The authors claim that this is the first time that multiple single copy genes were introduced into the pig germline. This is factually incorrect as several such pig lines have been produced for xenotransplantation e.g. using polycistronic vector constructs, PAC/Bac based constructs and multiple single transgenes placed in the porcine ROSA26 locus.

5) An extended breeding program was carried out resulting in 288 transgenic pigs, only a few of these were used for the experiments described. Why such a large number of animals? These cannot yet be used for food production.

6) The authors detected altered blood glucose levels; the authors should address potential causes, state whether this affects animal health?

7) The Introduction could be shortened and reduced to the most important facts.

8) In the Results section the authors should provide information about the cause for the death of the piglets. Out of 35 born Duroc piglets only 8 reached sexual maturity, why? The authors should also give the animal numbers of the piglets that survived, so that the reader could better track what happened to which pig and in which experiments was it used.

9) The pigs that harbored the Amp-resistance gene, did they serve as negative controls in Figure 2C,D, E? Did the authors find any differences between TG line 1 and 2? They should also give some information regarding the integration sites.

10) What role do Legumain and CEP112 play? Did they observe any adverse effects correlated to the integration site of the transgenes (only 6 piglets of line 2 survived)? Any implications the authors draw from their findings? They should discuss a targeted integration of the multi-transgene cassette into a safe harbor as described by Garrels et al., (2016) or Rieblinger et al., (2017).

11) Did the authors observe any differences in the meat composition of the transgenic pigs compared to wild type pigs? Do the authors expect any impact of the transgene expression on the microbiome of the pigs? The discussion part could be streamlined and reduced to the main findings and discussion points.

12) In the third paragraph of the Discussion section, were these data shown in the results? The authors should emphasize the importance of this pig model and should give some impressive calculations how much N and P could be saved worldwide.

13) The quality of some figures could be improved.

14) The authors should indicate from which pig line the figures are.

[Editors' note: further revisions were requested prior to acceptance, as described below.]

Thank you for resubmitting your work entitled "Novel transgenic pigs with enhanced growth and reduced environmental impact" for further consideration at *eLife*. Your revised article has been favorably evaluated by Ian Baldwin (Senior editor), a Reviewing editor, and three reviewers.

The manuscript has been improved but there are some smaller remaining issues that need to be addressed before acceptance, as outlined below:

Summary:

While it is acknowledged that the authors made a substantial effort to deal with the criticisms from the first round of reviews, a few issues remain to be solved prior to acceptance for publication. The authors have provided useful information in their rebuttal letter, but only parts of that have been integrated into the paper. The authors should carefully change that and add this information to the revised version of the paper.

Essential revisions:

Introduction: should read: “on soybean meals was decreased by.”

Introduction: should read: “have been established to date. Here, we established”

Subsection “Optimization and construction of a 2A-mediated salivary gland-specific multi-transgene”: Insertion of the transgene in a specific genomic locus can likely account for the lower expression efficiency (so called position effect).

Subsection “Generation of TG pigs”: this part is not easy to understand and needs clarification.

Subsection “Measurement of enzyme production in TG pigs”: should read “The results show that”

Subsection “Improved feed utilization and reduced nutrient emission in TG founders” should read “were significantly decreased in the TG pigs compared to that of the WT pigs. […] Fecal N and P excretion were decreased by”

Subsection “Enhanced growth performance in TG pigs”: the sentence is unclear and needs to be rephrased e.g. “To assess the growth performance, eight F1 TG pigs […] 50kg weights.”

Discussion section: should read: positive for the transgene.

Discussion section: “internal malformation in the digestive, circulatory or cardiopulmonary system. A previous study”

Discussion section: should read “in our study, the […] was in agreement with”

Discussion section: should read “high mortality rate, suggesting that the [...] may relate to the cloning technique.”

Discussion section: blank between “Kumar et al., 2013)” “The expression”

Subsection “Transgenic pigs”: add batch no. of the FBS, this is important when replicating the experiments.

---

## [Author Response]

Essential revisions:1) Although single cell clones positive for the transgene and EGFP were used for nuclear transfer, only 24 of the 35 piglets born were transgenic. The authors should discuss possible reasons why this happened.

The strict single-cell colony selection process usually includes a limiting dilution analysis to grow a single cell per well in a 96-well plate. This process is complicated and takes a long time to obtain cell colonies. As a primary cell, pig fetal fibroblast (PFF) used in transgenesis and nuclear transfer has a low-proliferating rate and limited lifespan. PFF would age and lose its proliferating activity when using a limiting dilution analysis. For PFF selection, we usually grew and selected them in a 10-cm dish in a higher cell density, as higher cell density facilitates keeping the proliferating activity of PFF. As a result, the obtained cell colonies were usually impure and included few cells from other colonies which may have different genotypes. We suppose this is the reason for obtaining transgene-negative pigs from a positive cell colony.

We have included these contents in the Discussion section. Some technical details were also added to Materials and methods section.

2) 8 of these pigs survived to sexual maturity. 4 piglets contained the plasmid backbone (ampicillin gene), which the authors claimed was due to breaks in the transgene. This could simply be a consequence of random transgene integration and not transposase mediated integration and the authors should adjust their explanation to reflect this likely possibility. The other pigs contained either a single copy integration in the Legumain or CEP112 gene or a triple integration in an intergenic region. It is unclear whether only four clones were used for nuclear transfer, and if so why these four clones were chosen. The authors should clarify these points.

Thanks for your helpful comments. We have adjusted the explanation in the related context to reflect this likely reason for the transgenic situation of the 5 piglets as suggested. For choosing cell clone/colony, we usually used cells with brighter fluorescence which implies a high expression of transgene. The chosen 4 cell clones had a higher EGFP expression than others. We have expanded the related context to describe it clearly.

3) The authors also need to give a brief description of the two genes (Legumain or CEP112) and if their expression was affected by the insertion of the transposon. Breeding was carried out with the line in which the transposon had integrated in CEP112, but no explanation is provided why this line was chosen.

Legumain is a cysteine protease that contributes to antigen processing for class II MHC presentation in lysosomes and can be overexpressed in some solid tumors. Previous report showed legumain knockout mice had a decreased weight gain (Shirahama-Noda et al., 2003). CEP112 gene encodes a centrosomal protein of 112 kDa which is localized around spindle poles, and its structure and function are still unclear (Jakobsen et al., 2011; Kumar et al., 2013). The pig lines in our study are heterozygous transgenic and the integration sites were intron region of Legumain or CEP112. Therefore, the expression of the two genes will not be affected significantly. No evident phenotypic change was observed in the two pig lines.

Transgenic line 1, 0707 and 0605 have one copy transgene inserted into intron 1 of Legumain, of which 0605 died at 2 days of age due to dysphagia (big tongue). 0707 was normal at birth but infected with pseudorabies virus at sexual maturity. Whereas line 2 (integration at CEP112) has 6 healthy pigs reach sexual maturity. This is why we chose line 2 for further breeding. We have added expanded text in the Discussion section of the revised manuscript.

4) In the feeding experiments the authors could show the functionality of the transgenes. Reduction of phosphorous output was however lower than pigs produced in 2001. The authors claim that this is the first time that multiple single copy genes were introduced into the pig germline. This is factually incorrect as several such pig lines have been produced for xenotransplantation e.g. using polycistronic vector constructs, PAC/Bac based constructs and multiple single transgenes placed in the porcine ROSA26 locus.

We have compared the 2 models in the Discussion section. Compared to the previously reported phytase TG pig lines that harbor a high copy number of a foreign gene in an insertion locus (2 to 35 copies of the transgene, producing 5 to 6,000 U/mL of phytase), our multi-TG pigs (1 copy transgene) secreted a lower amount of enzymes into the saliva. Our multi-TG pigs demonstrated similar digestibility with the high-expression phytase TG pigs, as well as similar reduced emission of P in manure. For grower-finisher pigs, our study displayed an approximately 45.7% reduction in fecal P when fed CS meals (70.3% phytates), versus 56% reportedly in high-expression phytase TG pigs fed on soybean meals (53% phytates).

We have deleted the description of “This is the first report that describes the integration of multiple single-copy genes into the pig genome” in the revised manuscript as suggested.

5) An extended breeding program was carried out resulting in 288 transgenic pigs, only a few of these were used for the experiments described. Why such a large number of animals? These cannot yet be used for food production.

According to the objectives set by “The National Major Project for Production of Transgenic Breeding Grant (2016ZX008006002) of China”, we need to produce more than 400 transgenic pigs as test/breeding materials. Besides the pigs used for the experiments in the manuscript, the rest were used for environmental and food safety assessment as well as determination of nutrient digestibility of different feeding stuff containing different levels of non-starch polysaccharide and phytic acid. All the transgenic pigs were under strict surveillance of the sponsor, The Ministry of Agriculture. The transgenic pigs cannot be used to produce food products until approval of the Ministry of Agriculture and must be treated harmlessly at the end of the experiments.

6) The authors detected altered blood glucose levels; the authors should address potential causes, state whether this affects animal health?

In cereal grains, arabinoxylans and β-glucans in the cell walls of starchy endosperm can act as barrier against nutrient hydrolysis and absorption. Our transgenic pigs can simultaneously express β-glucanase and xylanase which break the barrier of arabinoxylans and β-glucans in the cell walls, thus improve the digestion and absorption of carbohydrates. As a consequence, the blood glucose concentrations have a moderate increase accordingly.

The normal range of blood glucose is 4.3~8.6 mmol/L for grower pigs (Klem et al., 2010). The values between the transgenic and normal pigs were 4.97 vs 3.78 mmol/L in our study. Therefore, the changed glucose levels do not affect animal health significantly.

7) The Introduction could be shortened and reduced to the most important facts.

We have shortened the Introduction to give prominence to the key points.

8) In the Results section the authors should provide information about the cause for the death of the piglets. Out of 35 born Duroc piglets only 8 reached sexual maturity, why? The authors should also give the animal numbers of the piglets that survived, so that the reader could better track what happened to which pig and in which experiments was it used.

In the study, we produced a total of 33 live and 2 stillborn piglets. Approximately half (17) of the live cloned piglets died during the first month, mainly related to the visible malformation (30.3% of born alive), and weakness (18.2% of born alive) caused by various internal malformation in the digestive, circulatory or cardiopulmonary systems. Likewise, the littermate non-transgenic pig also died within 3 months after birth (4 pre-weaning and 3 post-weaning), suggesting the problem of high mortality rate of the pigs may focus on cloning technique per se. A high malformation rate (25.2%~39.8% of born) and mortality rate (48% of born or 58% of full-born) in transgenic and cloned piglets produced by SCNT were also found in previous studies (Liu et al., 2015; Schmidt et al., 2015).

There are a total of 16 surviving piglets after weaning. 13 out of them were transgenic, including 9 piggyBac-mediated and 4 randomly integrated transgenic pigs. 2/13 died from unknown disease after weaning. 3 randomly integrated transgenic pigs were excluded from further experiments for non-expression of transgenes. We have added more related description in the revised manuscript.

9) The pigs that harbored the Amp-resistance gene, did they serve as negative controls in Figure 2C,D, E? Did the authors find any differences between TG line 1 and 2? They should also give some information regarding the integration sites.

The pigs that harbored the Amp-resistance gene can normally expressed EGFP, but they harbored a defective upstream regulatory region of transgene. Thus, they failed to secrete transgenic enzymes in the saliva. They can serve as negative controls in Figure 2C,D, E.

The line 1 produced the same levels of salivary β-glucanase, xylanase, and phytase as that of line 2 (Figure 2). We did not find any differences between the 2 lines, including transgene expression levels, growth and reproduction performance. We have added the data of salivary β-glucanase, xylanase, and phytase activity of the TG (Figure 2 of manuscript).

10) What role do Legumain and CEP112 play? Did they observe any adverse effects correlated to the integration site of the transgenes (only 6 piglets of line 2 survived)? Any implications the authors draw from their findings? They should discuss a targeted integration of the multi-transgene cassette into a safe harbor as described by Garrels et al. (2016) or Rieblinger et al., (2017).

As explained in question 3, we have described the function of the 2 genes. The adverse effect of transgene integration is insignificant, as the transgene is heterozygous and integration site is intron. Furthermore, we were also trying to generate the transgenic pigs with inserted multi-transgene cassette into Rosa 26, a safe harbor locus. We currently have obtained this pig. Improved and expanded discussion has been added in the revised manuscript.

The high mortality in SCNT-derived pigs is quite normal in our laboratory, as we routinely produced many transgenic and gene-editing pigs. The high death rate of SCNT-derived pigs was also reported in previous publication (Schmidt et al., 2015).

11) Did the authors observe any differences in the meat composition of the transgenic pigs compared to wild type pigs? Do the authors expect any impact of the transgene expression on the microbiome of the pigs? The discussion part could be streamlined and reduced to the main findings and discussion points.

We have confirmed that the BgEgXyAp transgenes were specifically expressed in the parotid, submandibular, and sublingual glands, and its expression was undetectable in the muscle. Through testing the meat composition, no significant difference of meat compositions was found between TG and WT pigs except for calcium level, which was slightly lower in the TG pigs (Table 1).

For the gut microbiome, we did not observe the difference of composition of microorganism that output from the anus except for cyanobacteria (Table 2). The gut microbes of the TG pigs at different digestive tract and different feed nutrients need further investigation.

For the Discussion section, we have revised it as suggested.

Table 1. Analysis of the composition of porcine lean meat (% of DM）

ItemTG(n=3)WT(n=3)Pooled SD.P-valueCP%73.4975.933.400.45fat%21.1422.523.130.65Ash%4.973.151.720.41NFE%0.150.270.160.58NE (kJ/100g）2033.992131.5191.650.31Na (g/kg）1.361.270.060.25Mg (g/kg）0.810.790.070.88P (g/kg）9.017.681.120.30K (g/kg）12.0911.931.280.82Ca (g/kg）0.290.33*0.070.02Cr (mg/kg)0.270.180.090.06Mn (mg/kg)0.540.410.180.14Fe (mg/kg）56.9026.1229.710.11Cu (mg/kg)2.922.311.220.32Zn(mg/kg)58.0554.167.060.36Se (mg/kg)1.000.810.150.23Mo (µg/kg)108.5763.6850.490.18

Note: The average body weight, TG group: 102.06 ± 12.07 kg, WT group: 108.07 ± 10.10 kg

Table 2. Analysis of the composition of fecal microorganism of transgenic pigs fed HNSP diet

Taxon (%）WT(n=7)Tg (n=7）pBacteroidetes53.79 ± 19.6056.75 ± 8.110.72Firmicutes40.02 ± 17.0138.97 ± 8.070.88Spirochaetae3.76 ± 8.191.81 ± 2.170.56Proteobacteria1.53 ± 1.211.68 ± 0.710.77Cyanobacteria0.11 ± 0.150.38 ± 0.18*0.01Tenericutes0.05 ± 0.060.07 ± 0.040.34Lentisphaerae0.14 ± 0.180.02 ± 0.020.15Fibrobacteres0.40 ± 0.860.06 ± 0.070.38Others0.19 ± 0.130.25 ± 0.150.43

Note: Body weight of transgenic pigs is around 60kg.

12) In the third paragraph of the Discussion section, were these data shown in the results? The authors should emphasize the importance of this pig model and should give some impressive calculations how much N and P could be saved worldwide.

Yes, these data were shown in the Supplementary file 2. By our estimation, the transgenic pig of line 2 can save 26.8 kg diet per head (30~115 kg). Annual global pig production is approximately 1.2 billion heads. Overall, approximately 32.16 million tons feeding stuff (16% of crude protein and 0.6% of total Phosphorus) could be saved per year worldwide. Approximately 193.0 kiloton P and 823.3 kiloton N could be saved per year.

13) The quality of some figures could be improved.

We have improved the quality of figure 2, Figure 1—figure supplement 2, and Figure 1—figure supplement 5.

14) The authors should indicate from which pig line the figures are.

We have indicated pig lines in related figures or figure legends.

[Editors' note: further revisions were requested prior to acceptance, as described below.]

Essential revisions:Introduction: should read: “on soybean meals was decreased by”Introduction: should read: “have been established to date. Here, we established”

Above texts have been revised as reviewers’ comments.

Subsection “Optimization and construction of a 2A-mediated salivary gland-specific multi-transgene”: Insertion of the transgene in a specific genomic locus can likely account for the lower expression efficiency (so called position effect).

Thanks. This part described the expression of our vector in cells as an in-vitro validation of transgene. Therefore, it is not related to transgene insertion sites.

In transient transfection with Lipofectamine, the gene transfer capacity was affected by the size of plasmid DNA molecules, the level of transgene expression was inversely proportional to the length of the DNA molecule (Kreiss et al., 1999). Here, the size of four genes recombinant plasmid was larger than original monomeric counterpart. So, we think that the lower transfection efficiency of the large transgene construct likely accounted for the observed lower expression and enzymatic activity in porcine kidney (PK15) cells.

Subsection “Generation of TG pigs”: this part is not easy to understand and needs clarification.

Thanks for your helpful comments. Our genotyping results showed these founders’ transgene were inserted randomly, not by transposon. We have revised this sentence to “Among the 25 TG founders, 5 piglets (601, 603, 701, 705, and 709) harbored the fragments of the ampicillin-resistance gene of the transgene vector, implying the occurrence of a random, but not transposon-mediated transgene integration into host cell genome.”

Subsection “Measurement of enzyme production in TG pigs”: should read “The results show that”Subsection “Improved feed utilization and reduced nutrient emission in TG founders” should read “were significantly decreased in the TG pigs compared to that of the WT pigs. […] Fecal N and P excretion were decreased by”

Above texts have been revised in the manuscript.

Subsection “Enhanced growth performance in TG pigs”: the sentence is unclear and needs to be rephrased e.g. “To assess the growth performance, eight F1 TG pigs […] 50kg weights.”Discussion section: should read: positive for the transgene.Discussion section: “internal malformation in the digestive, circulatory or cardiopulmonary system. A previous study”Discussion section: should read “in our study, the […] was in agreement with”Discussion section: should read “high mortality rate, suggesting that the [...] may relate to the cloning technique.”Discussion section: blank between “Kumar et al., 2013)” “The expression”

Above texts have been revised as suggested.

Subsection “Transgenic pigs”: add batch no. of the FBS, this is important when replicating the experiments.

We have added batch no. of the FBS (Gibco 10100-147) and DMEM (Gibco 12491-023) in the revised manuscript.